# Application of Landscape-Ecological Approach for Greenways Planning in Rural Agricultural Landscape

**Jakub Melicher** [1,*] and **Jana Špulerová** [2]

1 Department of Ecology and Environmental Sciences, Faculty of Natural Sciences and Informatics, Constantine the Philosopher University in Nitra, Tr. A. Hlinku 1, 949 01 Nitra, Slovakia

2 Institute of Landscape Ecology, Slovak Academy of Sciences, P.O. Box 254, Štefániková 3, 814 99 Bratislava, Slovakia; jana.spulerova@savba.sk

* Correspondence: jakub.melicher@savba.sk

**Abstract:** This article presents an innovative approach to the concept of facilitating greenways into sustainable landscape planning. The greenways can be planned by application of a landscape-ecological concept, including analysis and synthesis of selected abiotic, biotic, and socio-economic landscape-ecological factors and recreation conditions, and by reviewing the current landscape structure and condition of linear components of green infrastructure. Determining the landscape ecological stability, visual impact of agricultural lands, potential erosion risks and real erosion processes, and identifying the natural, cultural, and historical values in the landscape, contributes to the design of ideal greenways placement and other linear components of green infrastructure. Applying these proposals to the agricultural landscape would increase the ecological stability and connectivity, decrease soil and water erosion risks, eliminate visual impact, and develop recreational infrastructure. In this way, greenways planning brings about a synergy between sustainable rural development, landscape and nature protection, and landscape aesthetics, which provides optimal landscape utilization and may encourage tourism and economic prosperity in the study area. Finally, in addition to the researched ecological benefits, our greenways proposal represents an alternative connection of settlements in rural agricultural landscapes, and so it can stimulate sustainable mobility and recreation as well as physical activity, health, and well-being.

**Keywords:** greenways; land use; landscape planning; sustainable mobility; recreation

## 1. Introduction

Recently, around 30% of agricultural landscapes in the EU have been facing a moderate risk of land abandonment in the context of an inter-related network of biophysical, farming, structural, market, regional, institutional, and policy factors that have affected decisions on land use and its changes. However, management issues and structural adaptation remain the key drivers of governance. Land use changes are the consequences of two main processes–intensification and marginalization, frequently seen together within the same area [1].

The rural regions of Slovakia are facing the same challenges of rural abandonment, largely as a result of abandonment or land use intensification [2]. Extensive land use has been substituted with intensive land use, which has brought about many environmental problems due to the sustainability gap. All local changes are happening against the background of global changes such as climate changes, biodiversity loss or deforestation. On the other hand, the rural landscape has much potential and many opportunities arising from its natural, cultural, and historical values, which could stimulate and strengthen rural development. It is the responsibility of rural regions to deal with these challenges and opportunities in a sustainable manner.

One of the best existing science-based tools that may provide sustainability in rural landscapes is landscape planning, which prescribes the optimal spatial configurations of

land uses [3]. Land use is the concrete reflection of human activities in time and space, representing a combined realisation of historical, economic, social, and cultural potential and interactions between natural values, technical opportunities, and human knowledge [4]. Integrating the landscape-ecological bases into existing planning and projecting processes could also be considered a critical tool of environmental management. This approach could be understood as the ecologization (greening) of spatial organisation, land use, and land protection [5].

Landscape ecology principles can solve land use issues by determining an appropriate landscape configuration. These principles provide a theoretical framework for evaluating, planning, and managing any greenway, given that a landscape manifests an ecological unity throughout its locality [6]. In addition to landscape ecological capabilities, there is also a close connection between the landscape and the attractiveness of the area which increases the demand for recreation. The landscape value consequently arises from the presence of important objects and the expression of cultural and historical heritage [7]. The ecological capabilities of landscape and landscape attractiveness are a well-suited framework for the overall purpose of greenways planning.

Greenways as linear components of landscape structure are also important elements of green infrastructure and represent bio-corridors which can be planned, designed and managed for multifunctional purposes [8,9]. They can increase ecosystem resilience, develop and maintain sustainability in rural communities, and support the mitigation of the impact of climate change [10]. We could consider greenways and all other linear elements of green infrastructure as an implementation of natural solutions for mitigating the impacts of climate change. In addition to natural benefits, they also provide other social and cultural services such as landscape perception, recreation, and wellbeing [11,12]. Greenways can be characterized in five keywords [13]: linearity, linking, multi-functionality, sustainable development, and spatial policies.

Salici [14] classified greenways into six different categories, according to the work performed by scientists and planners: urban riverside, recreational greenways, natural corridors of ecological importance, greenways of visual and historical value, greenways that aim to control the urban development, and comprehensive greenway systems and networks. According to Pena et al. [15] there are many approaches in Europe which focus on planning greenways and ecological networks according to different functions. The main functions are ecological, ecological stability, river systems, recreation, and landscape protection.

The ecological stability of a landscape can be identified as one of the essential problems of landscape ecology. In accordance with accredited ecological principles, species or associations are endangered if their stable functioning can be disturbed or disabled or if the conditions of their existence are unfavourable or their spatial isolation is expressed [16]. For conservation of the spatial ecological stability of the landscape, the designing of ecological networks, including bio-centres, bio-corridors, and interaction elements [17] is necessary.

The planning of greenways requires approaches which integrate abiotic, biotic, and cultural resources and issues [18]. The main focus of these approaches to greenways planning is to support biological diversity and to manage or direct urban development of recreational benefits. Determining the aim is a major step in planning and designing greenways [14].

One of the first studies related to greenways was published in 1976 by the Georgia Department of Natural Resources in the USA, on the topic of Environmental Corridor Study. It had four steps: (1) resource analysis, (2) corridor selection and priorities, (3) corridor planning and management options, (4) summary and conclusions [19,20].

According to Pena et al. [21], planning greenways in rural areas could provide various opportunities at many levels, such as protection and management of the natural and cultural heritages, mitigation of the socio-economic problems of the rural inhabitants, and expanding the options for including rural inhabitants in the development of the rural economy (and also cultural, recreational, and health benefits for people from city centres and for tourists coming from outside of the region). When targeting a rural landscape

where agricultural land use dominates the social-economic activity, the achievement of these various objectives (whether ecological, social, economic, developmental and cultural-communal) can be assisted by the use of greenways, which is a complementary concept to that of multifunctional agriculture as a tool for attaining these objectives [22]. The greenways planning approach in Slovakia and the Czech Republic is very close to traditional landscape planning concepts of landscape ecological planning and territorial systems of ecological stability. Both concepts share the common steps of analysis of abiotic, biotic and socio-economic factors, synthesis, evaluation, and propositions.

The aim of this article is to identify those methods of landscape ecological planning and capabilities of the landscape that can facilitate appropriate placement of greenways and other linear components of green infrastructure while maintaining the multifunctional purpose of greenways. The selected methods will then be applied to a study area in order to demonstrate their use in proposing greenways or other ecological network elements.

## 2. Materials and Methods

### 2.1. Methodological Approach

We adopted the methodology of landscape ecological planning for greenways planning, which consists of the following logical steps: (I.) analysis and synthesis, (II.) evaluations, and (III.) proposals [17,23]. For the first two of these steps, we selected landscape features to incorporate into our proposal for optimization of greenways routes in the landscape (Figure 1).

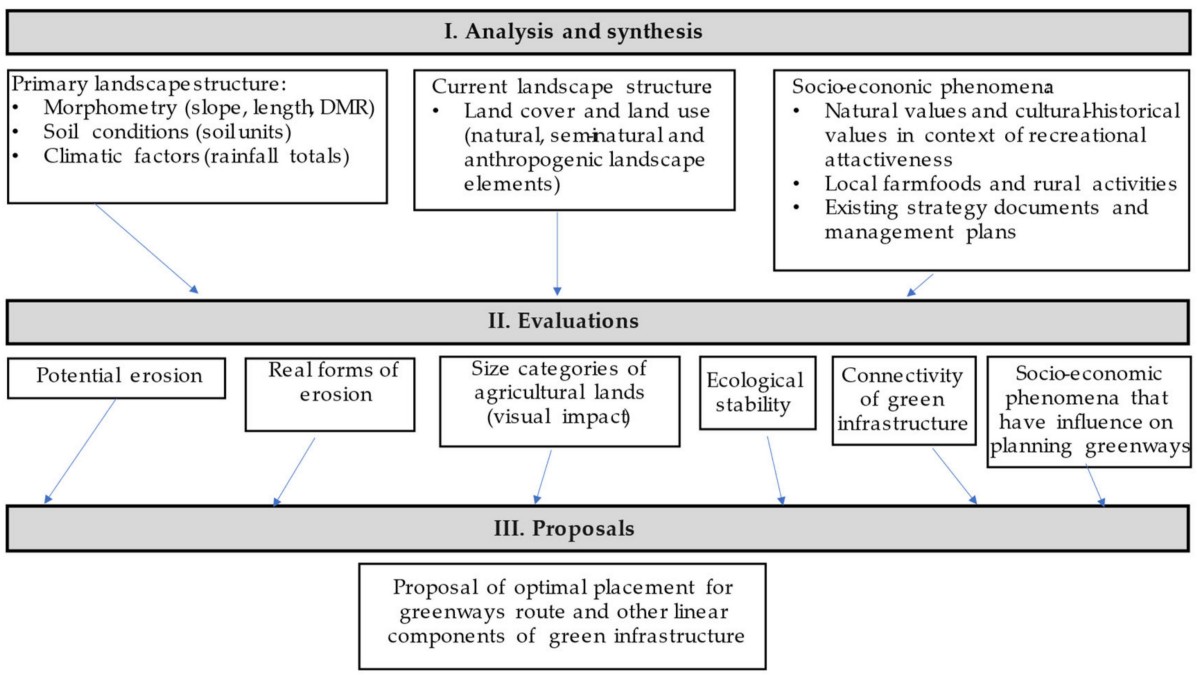

**Figure 1.** Application of logical steps for landscape ecological planning with selected attributes for greenways planning.

Input analytic data were obtained from various databases and available mapping sources. There is a need to focus more on agricultural landscape features, which have lower ecological stability and more visual impact. All analyses were created in ArcGIS 10.8. with the assistance of System for Automated Geoscientific Analyses software 7.8.2. to identify abiotic, biotic, and socio-economic conditions of the study area. The current landscape structure was created by data extraction and vectorisation of the most up-to-date orthophotomosaic of Slovakia, with additional details provided by field survey.

The goal of the Evaluation step is to determine the ecological quality of the landscape structure in its present state, and to define conflicts and pressures in the landscape. Evaluated indicators and data sources are listed in Table 1.

**Table 1.** List of selected indicators and evaluation methods.

| Evaluated Indicators | Input Data Description | Software | Used Tools | Source |
|---|---|---|---|---|
| Potential erosion | R factor, K factor, LS factor, where: | ArcMap 10.8 | Map algebra (R × K × LS factor) | Gallay [24], Midriak [25], Malíšek [26], Ilavská [27] |
| | R factor values were obtained from customised data from 6 different Slovak Hydrometeorological Institute ombrographic stations located around study area | | Interpolation (IDW) | Gallay [24], Malíšek [26], Ilavská [27], Slovak hydrometeorological institute (https://www.shmu.sk/sk/?page=1&id=klimat_zrazkosiet&region=BB; accessed on 10 December 2021.) |
| | K factor for main soil units of agricultural land resources, obtained from a map of soil ecological units (BPEJ) and from soil granularity data for forest land resources; data combined and classified by expert assessment (see cited sources) | | Georeferencing, Vectorisation, Polygon to raster | Gallay [24], Ilavská [27], Soil Science and Conservation Research Institute (https://portal.vupop.sk/portal/apps/webappviewer/index.html?id=d89cff7c70424117ae01ddba7499d3ad; accessed on 10 December 2021.), National forest centre (https://gis.nlcsk.org/arcgis/rest/services/Inspire/PodneTypy/MapServer; accessed on 10 December 2021.) |
| | LS factor–digital model of relief (created by aerial laser scanning) of study area (1 m × 1 m cell size) | SAGA 7.8.2 | Slope, aspect, curvature; Flow accumulation (Recursive); LS factor | Gallay [24], Geodesy, Cartography and Cadastre Authority of the Slovak Republic (https://zbgis.skgeodesy.sk/mkzbgis/sk/teren/export; accessed on 10 December 2021.) |
| Real forms of erosion | superposition of slope data derived from digital model of relief created by aerial laser scanning, and Orthophotomosaic of Slovak Republic; field survey | SAGA 7.8.2, ArcMap 10.8 | Slope, aspect, curvature (SAGA) | Geodesy, Cartography and Cadastre Authority of the Slovak Republic ( https://zbgis.skgeodesy.sk/mkzbgis/sk/teren/export; accessed on 10 December 2021.) |
| Size categories of agricultural lands (visual impact) | Boundaries and information about agricultural land were obtained from land parcel identification system (LPIS) and current landscape structure | ArcMap 10.8 | Classification of size categories of each agricultural plots | Soil Science and Conservation Research Institute (https://gsaa.mpsr.sk/2021/; accessed on 10 December 2021.) |
| * Current landscape structure | Analysis of Orthophotomosaic of Slovak Republic and field survey | | Vectorisation of each component of current landscape structure | Geodesy, Cartography and Cadastre Authority of the Slovak Republic (https://www.geoportal.sk/sk/sluzby/mapove-sluzby/wmts/wmts-zbgis.html/; accessed on 10 December 2021.) |
| Ecological stability | Analysis of current landscape structure and intensity of land use, field survey | ArcMap 10.8 | Classification of degree of ecological stability according to the authors | Reháčková and Pauditšová [28] |

**Table 1.** *Cont.*

| Evaluated Indicators | Input Data Description | Software | Used Tools | Source |
|---|---|---|---|---|
| Connectivity of green infrastructure | Analysis of current landscape structure and field survey | ArcMap 10.8 | Clip of forests, non-forest wood vegetation, wetlands, water surfaces, grasslands (pastures and meadows) and roads from current landscape structure | |
| Selected socio-economic phenomena that have influence on planning greenways | Superposition of selected socio-economic map data within study area | ArcMap 10.8 | Union of all identified socio-economic phenomena (listed below) | The monuments board of Slovak Republic (http://www.pamiatky.sk/po/po?Kraj=6&Okres=51&Obec=1620&KatastralneUzemie=2139&Ulica=&OrientacneCislo=&UnifikovanyNazovPO=&CUZText=&SearchButton=H%C4%BEada%C5%A5; accessed on 11 December 2021.) |
| | Cultural monuments | | | |
| | Protected areas, strategy documents and management plans | | | The State nature conservancy of Slovak Republic (http://maps.sopsr.sk/; http://www.sopsr.sk/ps.chvu2/files/Cerova-vrchovina-Porimavie.pdf; http://www.sopsr.sk/poprpokoradzskejazierka/ accessed on 11 December 2021.) |
| | Tourist routes | | | Hiking map (https://mapy.dennikn.sk/; accessed on 11 December 2021.) |
| | Panoramic views, outlook towers, mining cavities, farm food products, planned fruit tree gene fund | | | field survey |

* supportive analysis for evaluation.

The approach of Gallay [24] is to focus on particular landscape features that are identified as potentially important for defining water surface erosion risks, yielding the measure termed "potential erosion" [25]. This is erosion that could exist in the chosen area if there were no vegetation cover nor technical or biotic arrangements (i.e., it represents the maximum possible soil erosion which could occur). It is calculated from the values of abiotic coefficients by the equation: $E_{pot} = R \times K \times LS$, where $E_{pot}$ represents long-term average annual soil loss in tonnes per hectare per year, R represents rainfall erosivity, K represents soil erodibility factor and LS represents the slope-length gradient factor.

The results can be classified and interpreted as by Šúri et al. [29] as Erosion risk, and labelled according to the following scale:

1. ≤7.5 t/ha/year—no or low erosion risk from surface water erosion.
2. 7.6–22.5 t/ha/year—medium.
3. 22.6–75.0 t/ha/year—high.
4. 75.1–300.0 t/ha/year—ultra-high.
5. 300.1–900.0 t/ha/year—extremely-high.
6. ≥900.1 t/ha/year—catastrophic.

The real forms of erosion taking place in the agricultural landscape are determined visually from the orthophotomosaic, in combination with the gradient map derived from the digital model of relief (DMR) created by aerial laser scanning with cell size 1 m × 1 m. DMR helped us to identify erosion forms (expressed by high slope values) more accurately,

including in cases where woody vegetation is present. Erosion by rainwater flow is most visible in the forms of rills and gullies, which occur particularly in the agricultural landscape.

The current landscape structure and field blocks provided an adequate basis on which we could evaluate chosen attributes of land cover such as size categories of agricultural land, classified as microstructures (0.05–0.9 hectares); mesostructures (0.9–35 hectares); and macrostructures (35–100 hectares) [30]. The agricultural land in the study area consists of fields of arable land and grassland, each with specific crops. Macrostructures are homogenous surfaces in agricultural landscapes such as arable lands and have low ecological stability, notably higher soil erosion risks, and negative visual impact–they are detrimental to the aesthetic of landscape. Sanitation measures are proposed for macrostructures to delimit these areas to less than 25–35 ha. For comparison of historical and current land use, a historical and a current orthophoto map was used [https://mapy.tuzvo.sk/HOFM/; accessed on 11 December 2021].

The map of connectivity of green infrastructure was prepared from the current landscape structure, by selecting the forests, non-forest woody vegetation (both patches and lines), water surfaces and wetlands, grasslands, linear strips of grasslands, and roads that represent corridors surrounded by matrix [6].

For evaluation and classification of landscape ecological stability we customized a method of evaluation of degree of ecological stability based on the work of Reháčková and Pauditšová [28], where each element was classified by a degree of ecological stability (taking into account hemeroby as well), where 0 means low stability and 5 a high degree of stability. The assignment of degrees of ecological stability to elements of current landscape structure was performed on the basis of the criteria drawn up for each degree, such as the intensity of land-use of each element. The coefficient of ecological stability was calculated for current conditions and for the projected state of the landscape after implementation of our proposal. This results in numerical values which can be compared to indicate the effects of our proposals on the ecological stability of the area (Table 2).

**Table 2.** Interpretation of coefficient of ecological stability (CES) [28], reprinted from Eva Pauditšová(2007).

| Landscape Valuation | CES | Degree of Ecological Stability | Measures |
|---|---|---|---|
| landscape with very low ecological stability | 1.00–1.49 | 1 | high necessity of realisation of ecostabilization elements and ecostabilization management measures |
| landscape with low ecological stability | 1.50–2.49 | 2 | necessity of realisation of new ecostabilization elements and ecostability management measures |
| landscape with medium ecological stability | 2.50–3.49 | 3 | conditional necessity of realisation of new ecostabilization elements or application of adequate management measures |
| landscape with high ecological stability | 3.50–4.49 | 4 | realisation of adequate management measures |
| landscape with very high ecological stability | 4.50–5.00 | 5 | realisation of management measures to protect and maintain very high ecological stability |

The map of selected socio-economic phenomena that inform our planning of greenways and other linear components of green infrastructure was determined by superposition of the maps of the individual socio-economic phenomena, which are strongly related to recreation, tourism and rural development or nature protection, as determined from available open data sources and from relevant institutions and authorities. Evaluation of selected socio-economic phenomena included visual evaluation, which helped to make decisions of greenways route placement that would maximise their recreational potential and introduce linear elements of green infrastructure into local bio-corridors already existing within the territorial system of ecological stability.

In this paper, we apply the stated measures for achieving our goals (i.e., the design of greenways and linear components of green infrastructure) to generate concrete proposals for a particular study area. For this study, it was necessary to take particular note of attributes of the landscape which impact the proposals for planning greenways according to the multi-use definition of greenways [8,9] aimed at elimination of pressures and negative impacts of evaluated indicators (Table 1). The difference between greenways and other linear elements of green infrastructure is that greenways have multi-use functions, so their study and management requires consideration of their ecological, delimitative (interactive), ecostabilization, greening, aesthetical, social, and economic functions, and of ways to link settlements with attractive areas of the landscape. Other linear components of green infrastructure have more minor functionality and are not included in our proposal primarily for social and economic reasons, but in order to support landscape connectivity (migration of species), delimit macrostructures (decreasing of visual impact), reduce potential erosion risks and stabilise real forms of erosion.

### 2.2. Study Area

The study site lies in the south of Central Slovakia and consists of two Cadastral areas–Nižná Pokoradz and Vyšná Pokoradz (Figure 2, Table 3). They are rural communities situated north of the city of Rimavská Sobota. They have an agricultural landscape character, and the closeness of the city does not affect the village character. Cultivated fields and distant forests surround both villages. In terms of land use, the study area consists of agricultural landscape (51%), forest landscape (46%), and urbanised landscape (3%).

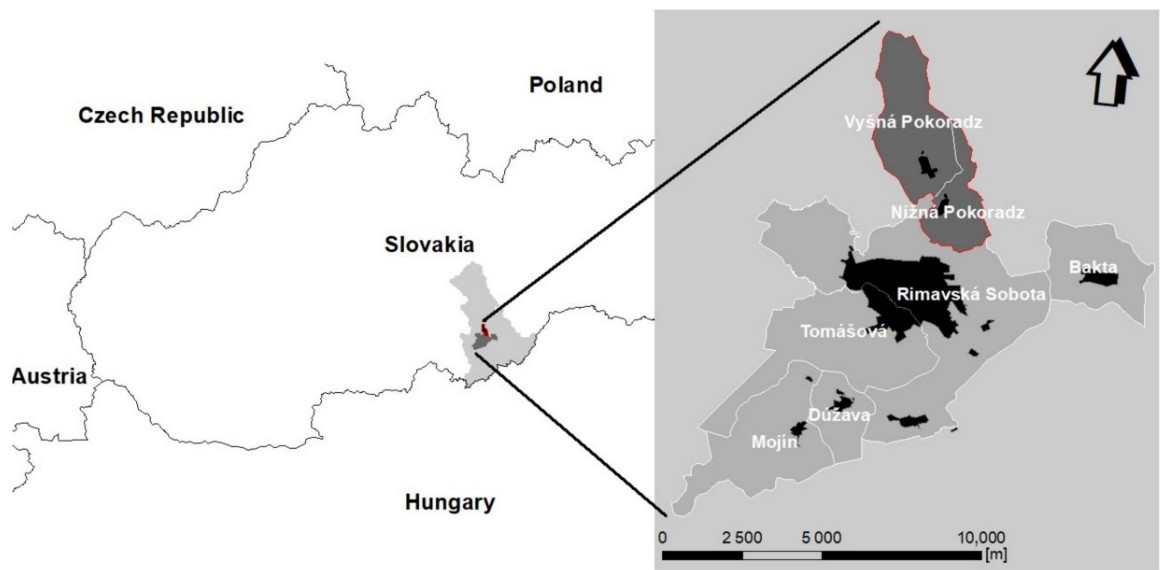

**Figure 2.** Location of study area–Rimavská Sobota district (grey), Rimavská Sobota town municipality, town communities (white outlines) and study area (red outlines).

**Table 3.** Basic characteristics of the two villages in the study area.

|  | Vyšná Pokoradz | Nižná Pokoradz |
|---|---|---|
| Population (as of 20. 3. 2017) | 252 | 250 |
| Cadastral area [ha] | 902 | 408 |
| Altitude above sea level [m] | 370 | 334 |

Arable lands predominate in the agricultural landscape over grasslands, the latter of which consist mainly of hay meadows (including the habitat of European interest 6510, Lowland hay meadows), pastures, and Steppic enclaves (frequently constituting the

habitat of European interest 6240, Sub-Pannonic Steppic grasslands). There are also other landscape structural elements such as gardens and orchards, which are integrated into gardens, allotments, and mosaics, or are overgrown with non-forest woody vegetation. Special attention needs to be paid to the presence of some wetland elements (swamps, periodic swamps, and especially habitats of European interest 3150 Natural eutrophic lakes with *Magnopotamion* or *Hydrocharition*).

The forest landscape is made of forest and non-forest woody vegetation. Most of the forests in the study area are a part of the Site of Community Importance Poko-radzské Jazierka, containing a habitat of European interest (91H0*-Pannonian woods with *Quercus pubescens*), and a habitat of national importance (Sub-continental *Quercus-Carpinus betulus* forests). Some parts of the forest have an open-forest character due to abandoned wood pastures and rocky slope soil conditions, or due to being located on gullies. Non-forest wood vegetation has a point, diffusive, linear, or patch character and is localised on grasslands and arable lands. Both forests and non-forest woody vegetation created by trees and shrubs provide important bird habitats, designated as the Special Protected Area Cerová Vrchovina-Porimavie (Birds Directive).

The altitude is between 249.5 m and 523.3 m above sea level. In terms of gradient, the study area consists of slopes 3–7° (37% of study area), 0–3° slopes (29% of area), 7–12° slope (18% of area) and 12–25° slope (14% of area). This is due to the study area's location on the north edge of the Southern Slovak Basin, in contact with volcanic parts of the Revúcka Highlands.

## 3. Results

### 3.1. Analysis of Current Landscape Structure and Size Categories of Agricultural Land

The analysis of the current landscape structure (Figure 3, Table 4) is the principal basis for the next evaluation.

**Table 4.** Analysis of current landscape structure.

| Elements of Current Landscape Structure | Area (ha) | Share (%) |
|---|---|---|
| 1.1 Forests | 526.46 | 40.23 |
| 2.1 Non-forest woody vegetation (NFWV) | 69.80 | 5.33 |
| 3.1 Hay meadows | 158.05 | 12.8 |
| 3.2 Pastures | 11.10 | 0.85 |
| 3.3 Grasslands with NFWV < 25 % | 38.47 | 2.94 |
| 3.4 Grasslands with NFWV < 50 % | 12.87 | 0.98 |
| 4.1 Arable land–small fields | 3.18 | 0.24 |
| 4.2 Arable land–large blocks | 380.00 | 29.4 |
| 4.3 Gardens | 19.49 | 1.49 |
| 4.4 Mosaics of small fields | 34.32 | 2.62 |
| 5.1 Natural water surfaces | 2.53 | 0.19 |
| 5.2 Occasionally-flooded areas | 3.00 | 0.23 |
| 6.1 Natural rocky elements | 0.97 | 0.07 |
| 6.2 Abandoned quarry | 0.36 | 0.03 |
| 7.1 Photovoltaic power station | 2.85 | 0.22 |
| 8.1 Farm complexes | 4.27 | 0.33 |
| 8.2 Dung yard | 0.17 | 0.01 |
| 9.1 Settlements | 15.41 | 1.18 |
| 9.2 Graveyards | 1.87 | 0.14 |

**Table 4.** *Cont.*

| Elements of Current Landscape Structure | Area (ha) | Share (%) |
|---|---|---|
| 9.3 Allotments | 16.65 | 1.27 |
| 10.2 Metalled roads | 6.80 | 0.52 |
| Total: | 1308.61 | 100 |

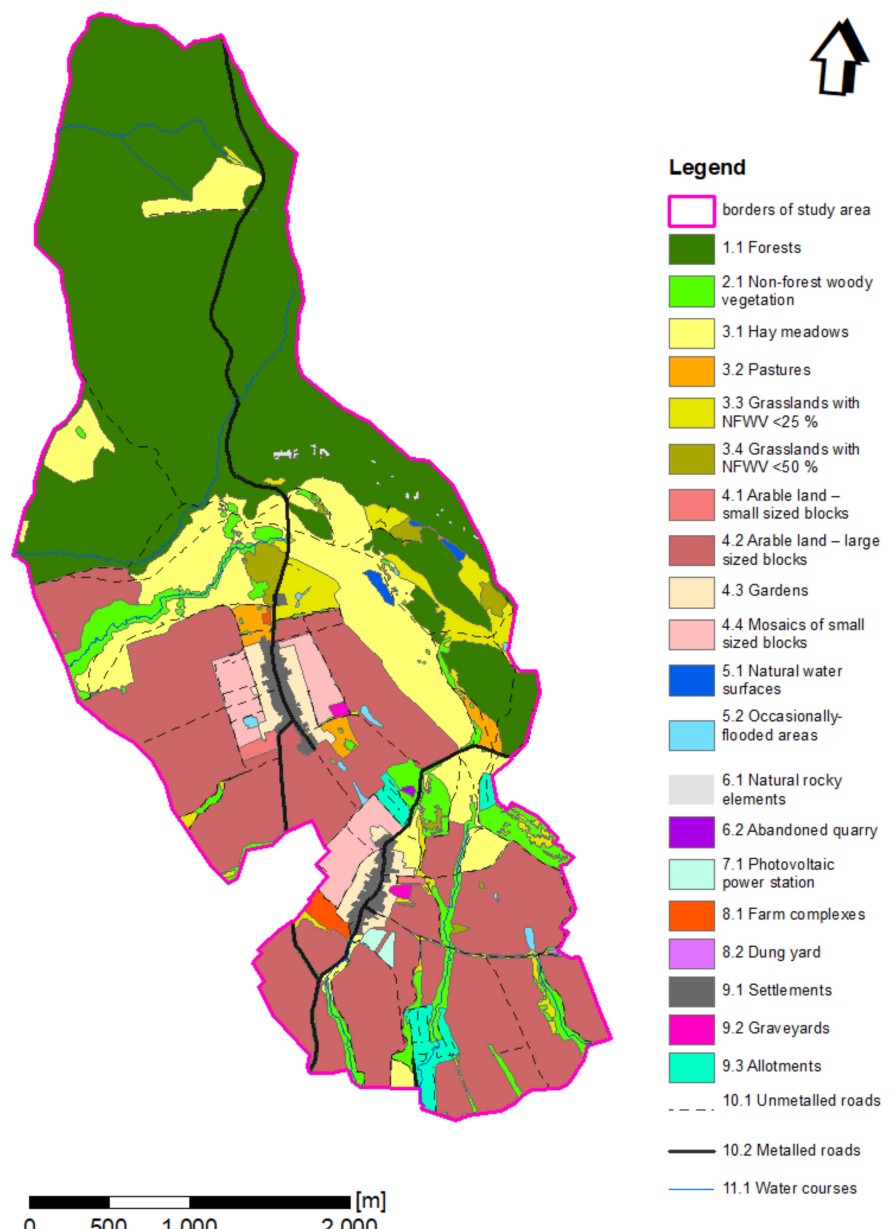

**Figure 3.** Map of current landscape structure.

Around one third of the agricultural landscape is created by macrostructures (plots with more than 35 hectares). These structures are present as negative elements in the landscape, since they decrease the heterogeneity of landscape structures and create negative visual impact. These macrostructures are relatively recent in origin, as they were formed after the 1950s during collectivization. Our recommendations here would be delimitation of these landscape structures to small-size blocks, and filling the gaps between blocks by ecological network elements such as trees, shrubs and roads, which are also integral parts of

greenways. The relict mosaics of traditional small-sized blocks with arable land, grassland, and pastures of fruit groves are well preserved around the villages (Figure 4).

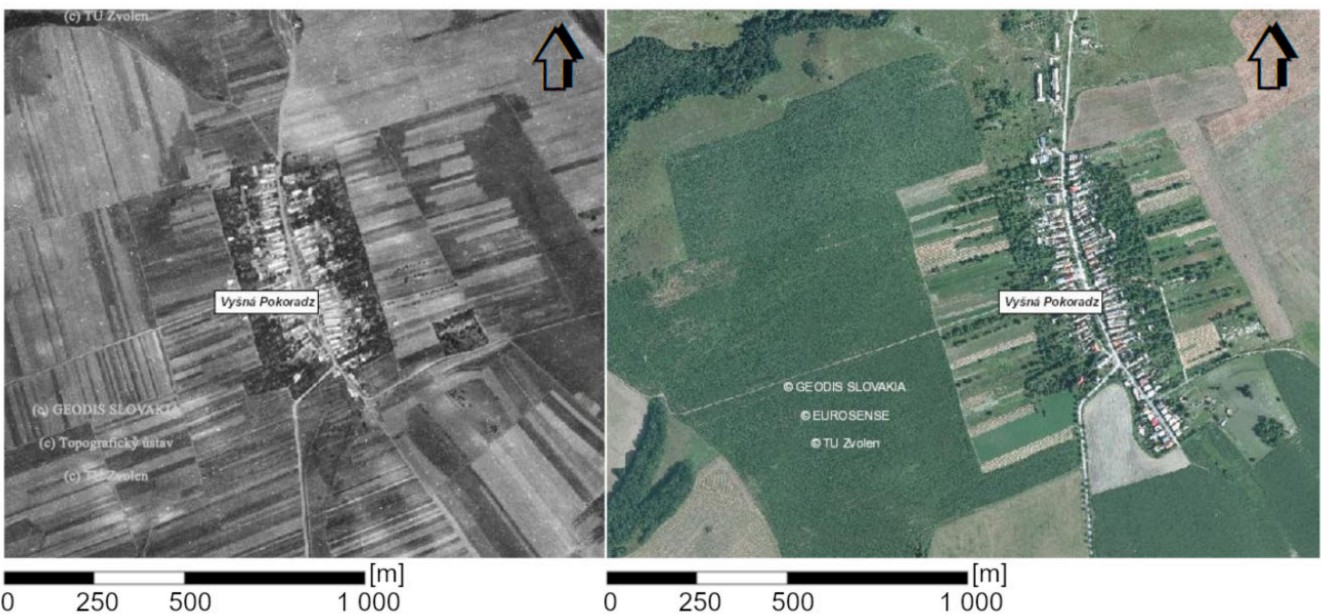

**Figure 4.** Comparison of land use of the agricultural landscape surrounding the village Vyšná Pokoradz in 1950 (on the **left**) and 2010 (on the **right**).

### 3.2. Ecological Stability and Connectivity of Green Infrastructure

The coefficient of ecological stability is 2.98 for the study area as a whole; this is a result of the relative occurrence of comparatively stable patches (woods, non-forest woody vegetation, extensively used grasslands, wetlands and natural water surfaces) and comparatively unstable patches (settlements, roads, arable lands). The map of ecological stability shows the location of unstable patches, which would benefit from the placement of ecostabilization measures such as greenways and other linear elements of green infrastructure into the landscape.

The connectivity of green infrastructure is another important aspect of the landscape structure, which contains linear corridors, some of which connect patches of non-forest woody vegetation core areas. Most connectivity is provided by non-forest woody vegetation in channels of watercourses. The patches thus connected are areas of forests, grasslands (hay meadows, pastures) or allotments, which are seldom or only periodically inhabited.

### 3.3. Potential Erosion and Real Forms of Erosion

We distinguished four categories of erosion risk based on GIS synthesis and evaluation of potential erosion: (Figure 5):

1.  No or low soil erosion risk—almost the whole area (0–7.5 tons per hectare per year).
2.  Medium soil erosion risk—predominantly on arable lands (7.51–22.5 tons per hectare per year).
3.  High soil erosion risk—mainly in forests, partially on agricultural landscape and around watercourses (22.51–75 tons per hectare per year).
4.  Ultra-high soil erosion risk—predominantly in forests, rocky slopes and in gullies (75.01–300 tons per hectare per year).

Most of the areas with high and ultra-high erosion risk are ecologically stabilized by the presence of xerothermic vegetation, forests, and non-forest woody vegetation, especially in channels of watercourses deeply sliced into the terrain (where gully erosion is expressed the most). These places are not ideal for greenways due to the slope and higher erosion risks. Other areas of arable lands are not protected from erosion and are less stable. To

decrease soil loss on arable lands, it is necessary to apply ecostabilization measures, but in the context of the current topic we propose only the planting of linear ecostabilization elements such as linear woody vegetation (infiltration belts of shrubs and trees), preferably along elevation contour lines.

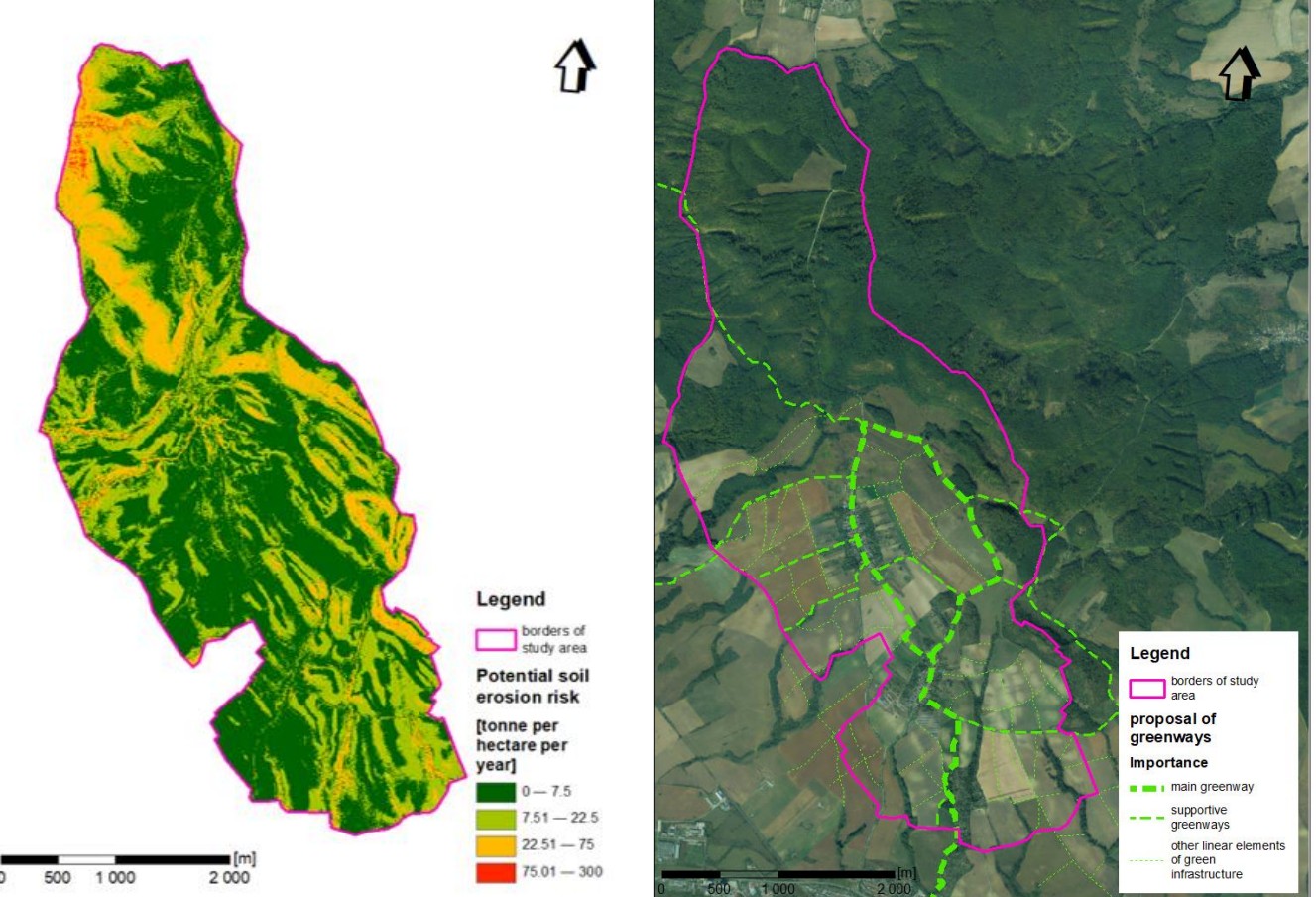

**Figure 5.** Map of potential soil erosion risk (**left**) and current orthophotomosaic with proposed greenways (**right**).

The rill erosion forms were identified mostly on valley lines (Figure 6). For sanative measures, which represent a form of eco-stabilisation, greening (i.e., planting of grass culture, or even woody vegetation) is appropriate. It can stop or reduce the erosion processes, and has other benefits as well.

### 3.4. Selected Socio-Economic Phenomena That Inform Planning of Greenways and Other Linear Components

Protected areas, including the Site of Community Importance (SCI) Pokoradzské Jazierka, which is also a nature reserve, and the Special Protection Area (SPA) Cerová Vrchovina–Porimavie, were identified from our analysis of socio-economic features as being attractive for recreation (Figure 7). The Pokoradzské Jazierka site is also integrated into the territorial system of ecological stability as regional bio-centre. The SCI (SKUEV0364) Pokoradzské Jazierka, with a total surface area of 62.24 hectares, was designated within existing nature reserve areas for protection of habitats of European interest, which are mostly xerothermic forests with oaks, hay meadows, pastures, wetlands, and rocky slopes. Several beetle species of European interest are present there. Another important part of the study area is that of the SPA Cerová Vrchovina-Porimavie (SKCHVÚ003), designated under the Bird Directive for securing favourable status of bird species habitats of European

interest and habitats of species. The management plan of SPA Cerová Vrchovina-Porimavie was developed in 2015 to protect these species.

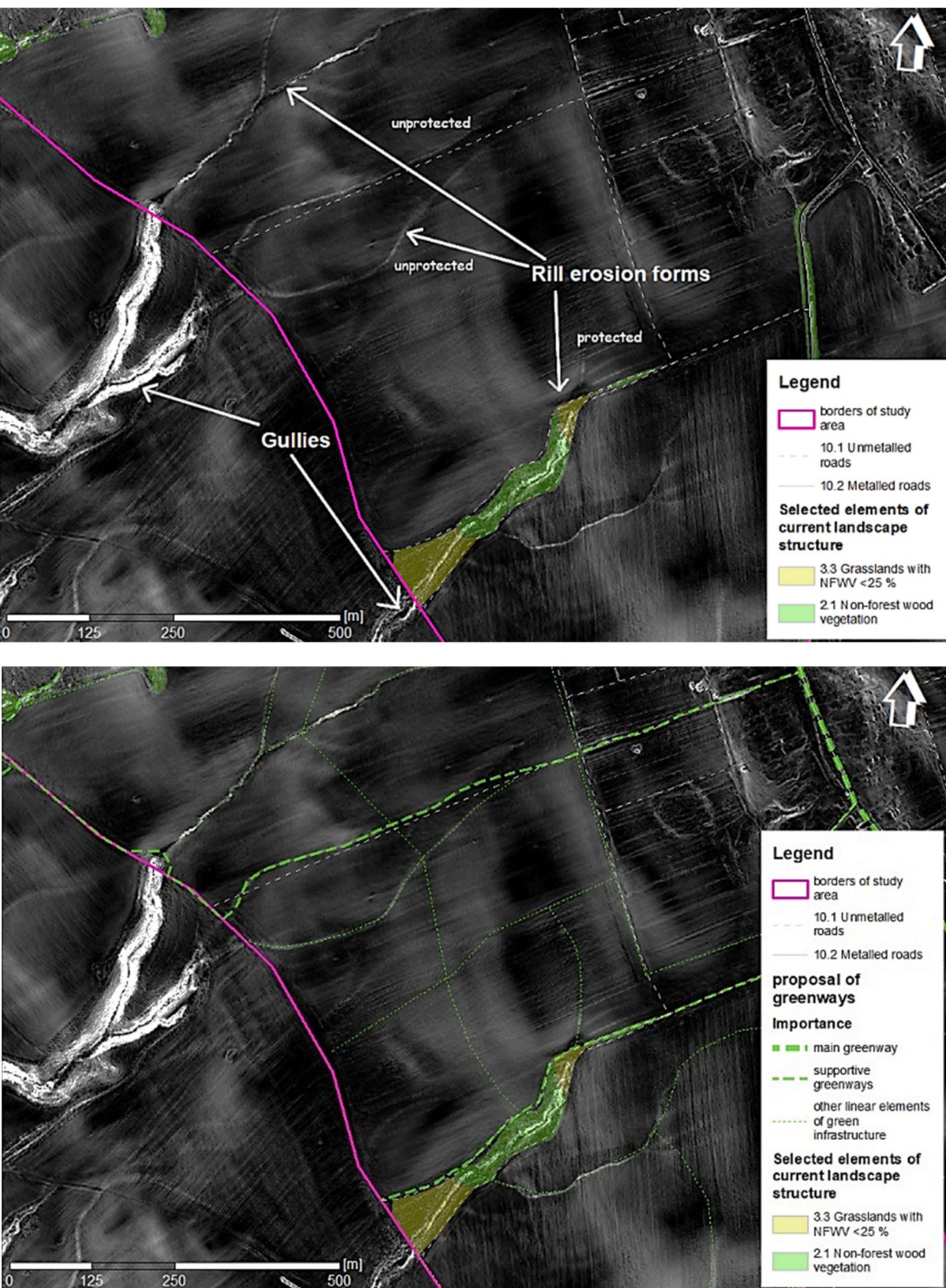

**Figure 6.** Identified forms of real erosion on slope map–current situation (**above**); proposal for greenways and other linear elements of green infrastructure as sanative measures (**below**).

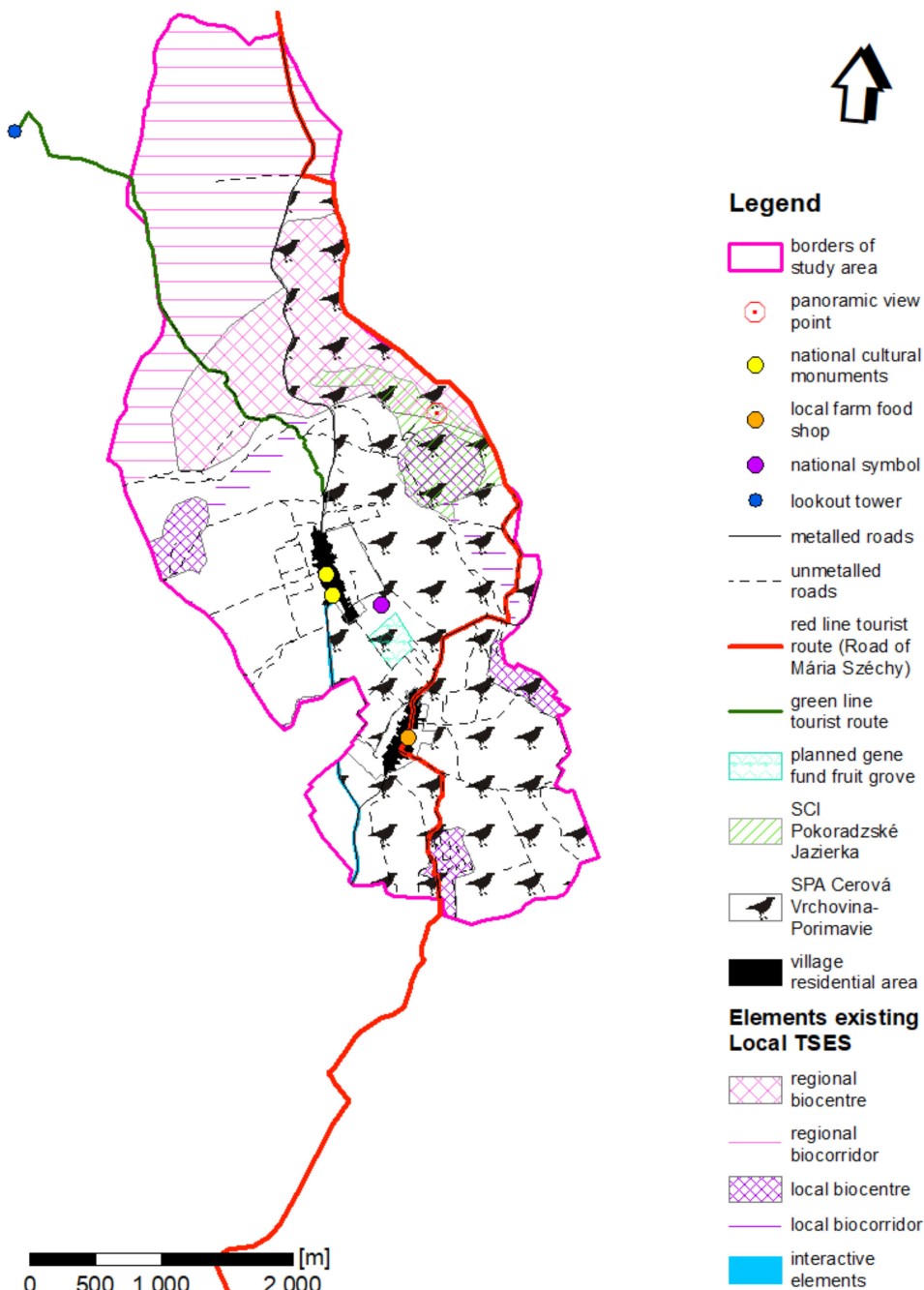

**Figure 7.** Map of selected socio-economic phenomena that have an influence on the planning of greenways and other linear components.

The most popular locality for recreational use in the study area is Kamenný Janko (literally, "Stoney Johnny"), which is one of the highest points in the area and has a nice panoramic view. It consists of a volcanic andesite neck and there are amazing rocky elements similar to earth pyramids. Its xerothermic rocky slopes support endangered plant species. Last but not least, there are attractive "caves", but these are man-made mining cavities, whence the material was mined in the past for the construction of houses and roads. These cavities show the profile of different geological layers and are an important habitat for bats.

The recreational infrastructure also includes two tourist routes (the red line linking Fiľakovo Castle and Muráň Castle passes above the Kamenný Janko panoramic viewpoint; the green line links Vyšná Pokoradz village with the lookout tower Maginhrad), and two

national cultural monuments–a classic bell-tower and a Protestant church. There is also a monument consisting of the national symbol (a Slovak double-barred cross) on the hill, where sheep graze. In the context of rural development, it is necessary to promote local farm products, which consist of one local farmer and the presence of a gene fund of orchard trees managed by a local NGO. The whole area has recreational, scientific, and educational value and the potential for rural development and agrotourism development in this region.

There are no tourist routes for non-motorized crossings of pedestrians and cyclists which could connect Rimavská Sobota town with the localities Kamenný Janko and Pokoradzské Jazierka. At present there are only indirect routes through field and forest pathways. There is a certain amount of recreational land use at present. However, it would be a good idea to modify and manage tourist routes in order to increase capacity, connectivity, and orientation. In addition, it would be desirable to add such recreational equipment as shelters, maps and informative or educational panels.

A study of the local territorial system of ecological stability of Rimavská Sobota [31] has identified the location of regional and local bio-centres, bio-corridors, and interactive elements, with an eye towards increasing the ecological stability and connectivity of the area.

### 3.5. Proposal of Greenways

The analysis of natural and socio-economic phenomena, and synthesis of the results, are crucial for proposing alternative routes for greenways, which have multi-use functions that go beyond merely acting as bio-corridors. The result of this process is a comprehensive map, which assists us in making decisions for alternative greenway routes. The location of proposed greenways is informed by landscape-ecological elements and phenomena (non-forest woody vegetation, forests, extensive land use of grassland, elements of territorial systems of ecological stability, protected areas) and landscape-aesthetical objects which are used for recreation or have importance from the perspective of recreation development. In our study area, we identified certain natural and cultural elements of the landscape as valuable: mosaic structures, mining cavities, symbolic places, and meaningful elements of the current landscape structure (andesite necks, earth pyramids). Many of these objects are part of SCI Pokoradzské Jazierka.

The landscape in the area is under pressure from potential soil erosion risks, disruption of connectivity, and a combination of various negative factors such as low ecological stability and low visual impact (on the macro- and mesostructural level). During our field survey, we found that the tourist routes were overgrown or ploughed.

Our greenways proposal has three parts, which are (Figure 8):

- a main greenway route, containing a loop which connects the two settlements with SCI Pokoradzské Jazierka; the route also extends to the urbanised area of town Rimavská Sobota as well. This is intended to serve a multi-use purpose. The total length of the proposed route is 7.68 km.
- supportive greenways to link nearby settlements with the study area, or provide shorter routes between nodes. These should also be multi-use, but mainly support connectivity to the main greenway route. The total length of the proposed routes is 15.58 km.
- other linear elements of green infrastructure (delimitation or interactive elements)–this measure is aimed at macrostructures and areas of potential soil erosion risks, real erosion forms, low connectivity or low ecological stability. The total length is 31.82 km.

It is also necessary to consult stakeholders and modify the greenways plan accordingly, as they are key players in rural development and important for the decision-making process. The implementation of greenways also includes indispensable administrative procedures, land use consolidations, road adjustment and planting of appropriate wood species (Figure 9). Planting must respect also agrotechnical practice. All types of greenways constitute ecostabilization measures.

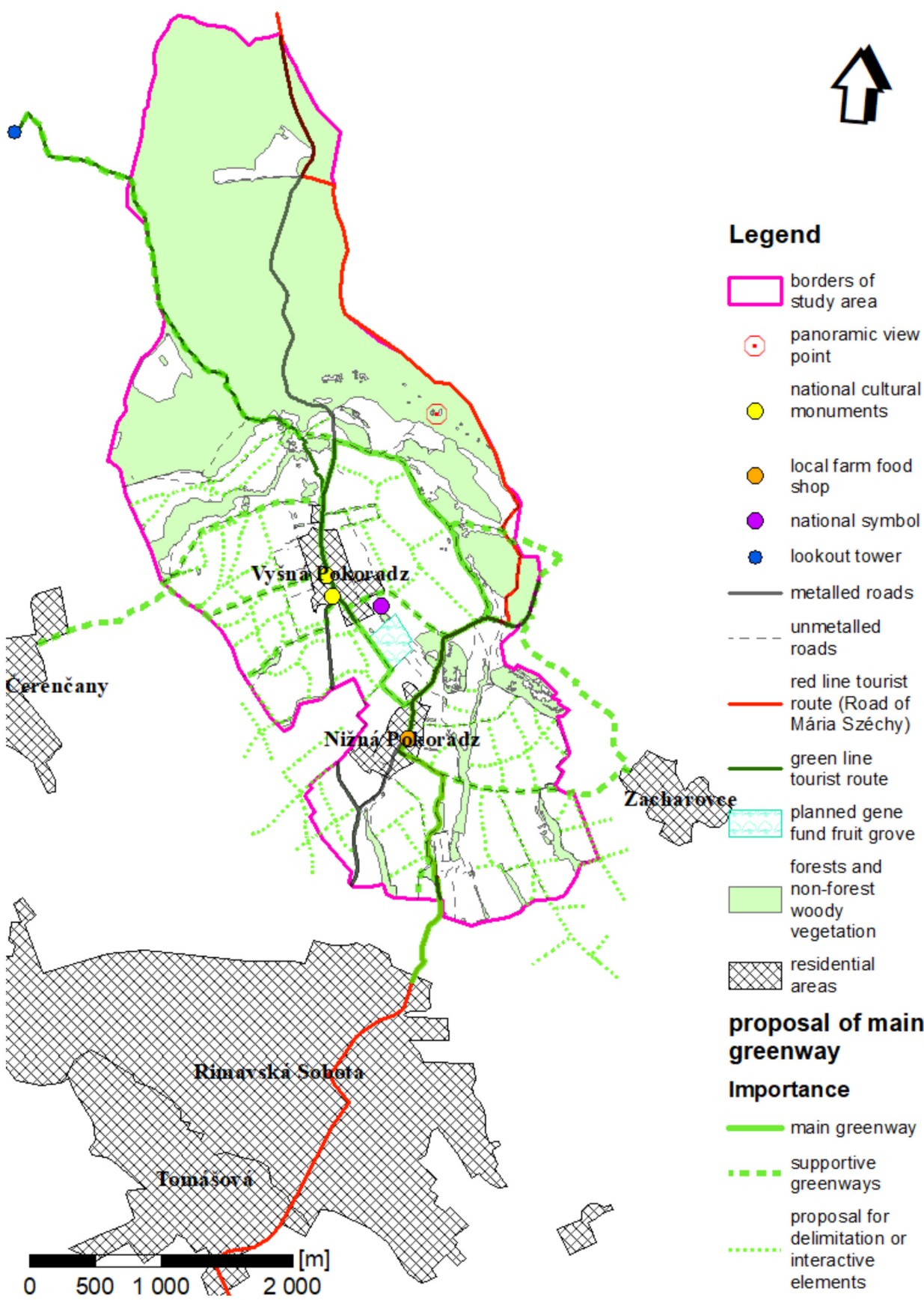

**Figure 8.** Map of proposal of greenways and linear components.

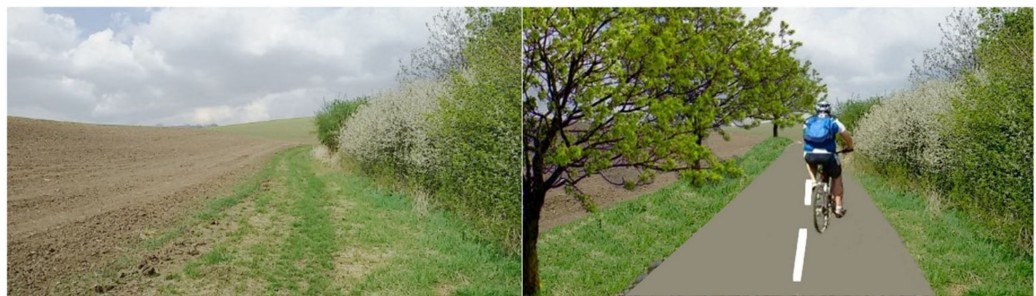

**Figure 9.** Potential route visualization: existing red link tourist route (on **left**), visualization of greenways proposal (on **right**).

To show the effect of the proposed greenways and other linear elements on the landscape's capabilities, we shall compare the current situation with that which would obtain if our proposed greenways were implemented.

To evaluate the potential contribution of greenways, we use the maximum number of greenways and other linear components of green infrastructure in order to decrease the size of macrostructures (Figure 10). The design of these components is informed by the results of erosion risk evaluations, as they are intended to have a synergetic effect on the delimitation of macrostructures. If this proposal were applied, it would totally eliminate the macrostructures as well as their visual impact. Elimination of macrostructures would also affect the total area of mesostructures, whose total surface would increase from 448.41 ha (72%) to 590.37 ha (94.8%). The impact on microstructures would be small, increasing the area from 28.24 ha (4.5%) to 32.34 ha (5.2%).

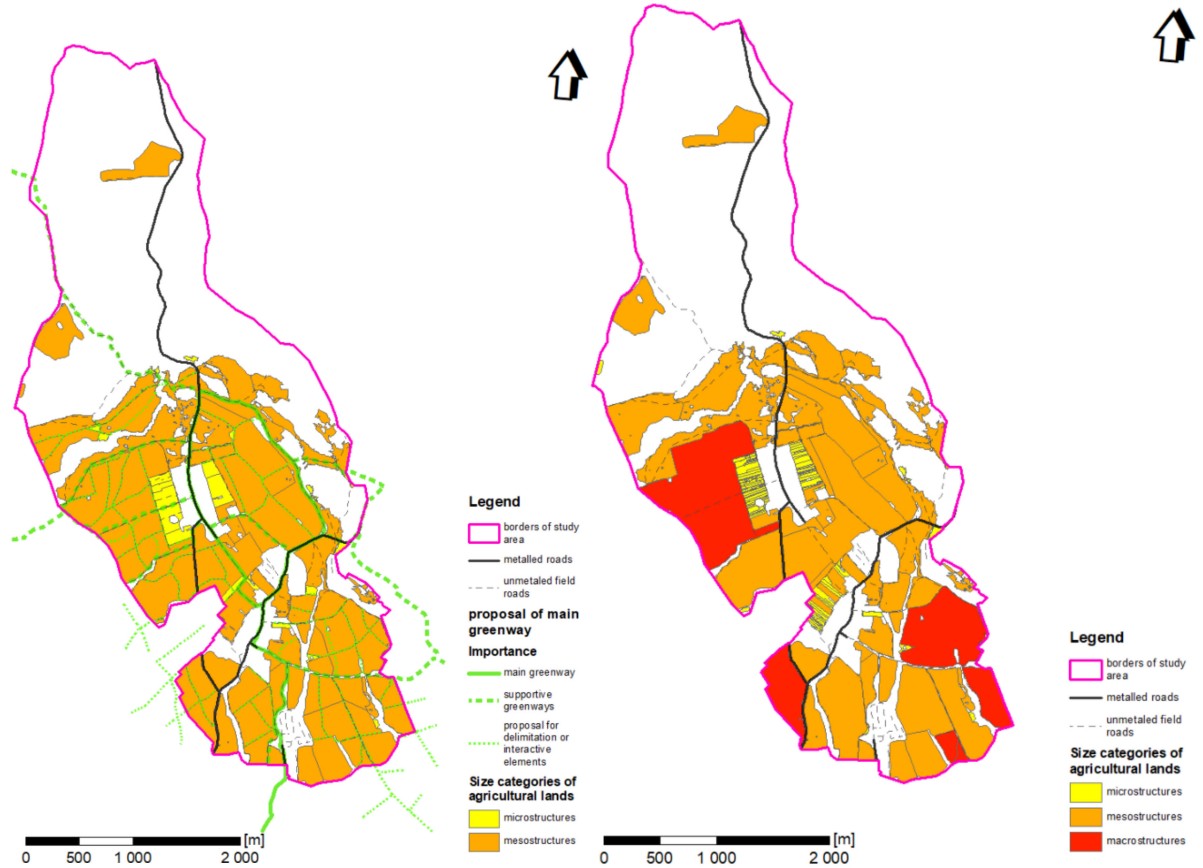

**Figure 10.** Map of size categories of agricultural lands–current (on **left**); after application of proposal for delimitation or interactive elements (on **right**).

We also calculated the coefficient of ecological stability for the model of the proposed greenways (except for roads and settlement areas) and other linear components, with a buffer zone of 10 m from the central line of the component (Figure 11). In our calculation we assigned the third degree of ecological stability (medium ecological stability) to the proposed elements; this is also the value of non-forest woody vegetation (linear). The proposed model has a coefficient of ecological stability of 3.02 (+0.04 increase), which expresses the conditional necessity of realisation of new ecostabilization elements or application of adequate management measures, which can increase ecological stability. There is a need for further measures targeted at the agricultural landscape, particularly arable lands, which have the lowest ecological stability and disturb the connectivity of patches with higher ecological stability. Taken together, the proposed linear elements of the presented model (which would have a total area of approx. 38 ha) did not affect the value of the coefficient much, but they provide better connectivity of patches. To increase the coefficient, it is necessary to consider other land use changes, ideally extensive use of grasslands or using grass to delimit parts of arable lands.

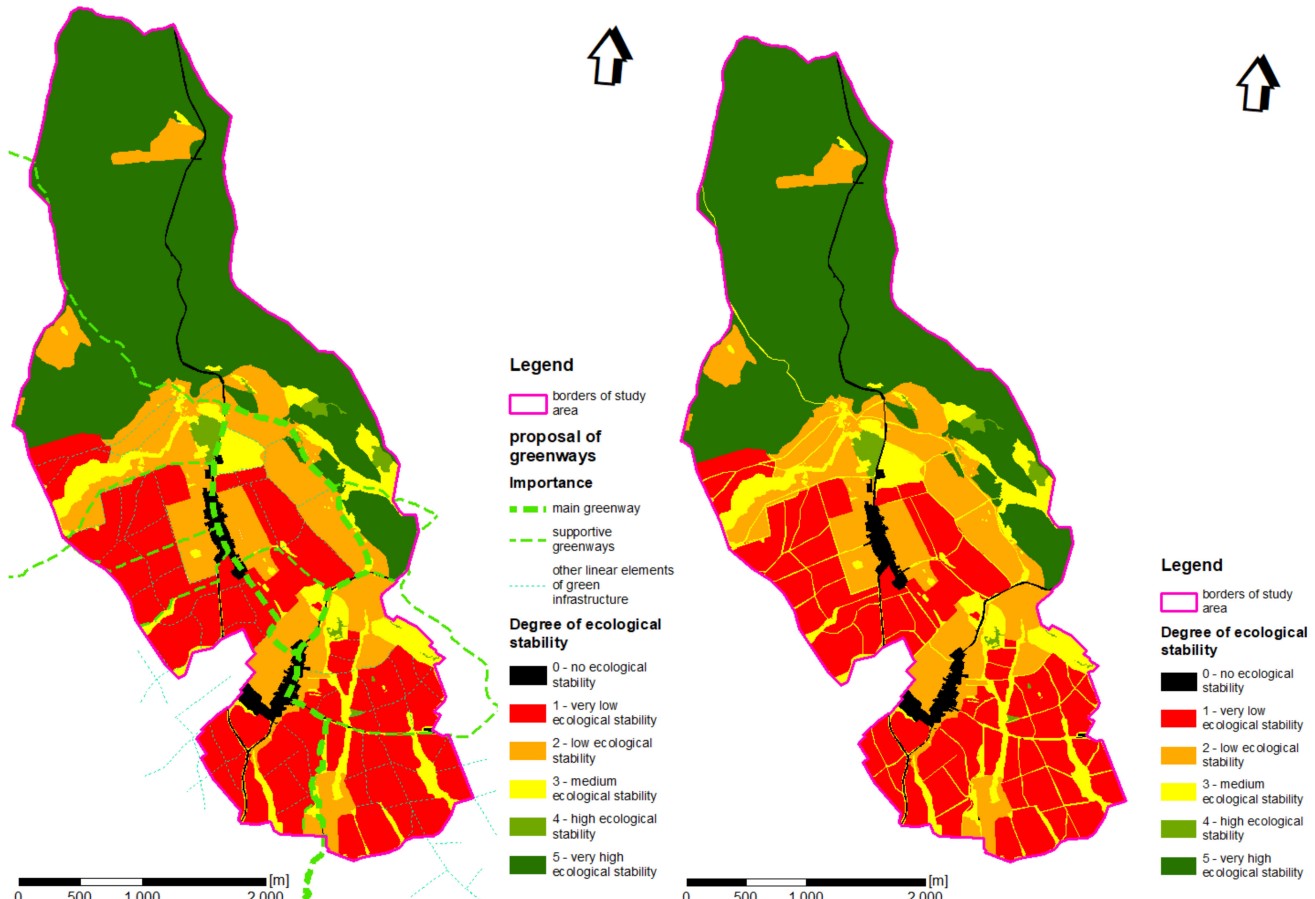

**Figure 11.** Map of ecological stability–current, with planned greenways and other linear elements shown (on **left**); after application of our proposals (on **right**).

We also used our proposal of green infrastructure for appraisal of landscape connectivity (Figure 12). Such an analysis also assists in decision-making that takes into account important ecological factors such as migration and defragmentation of stable parts of the landscape.

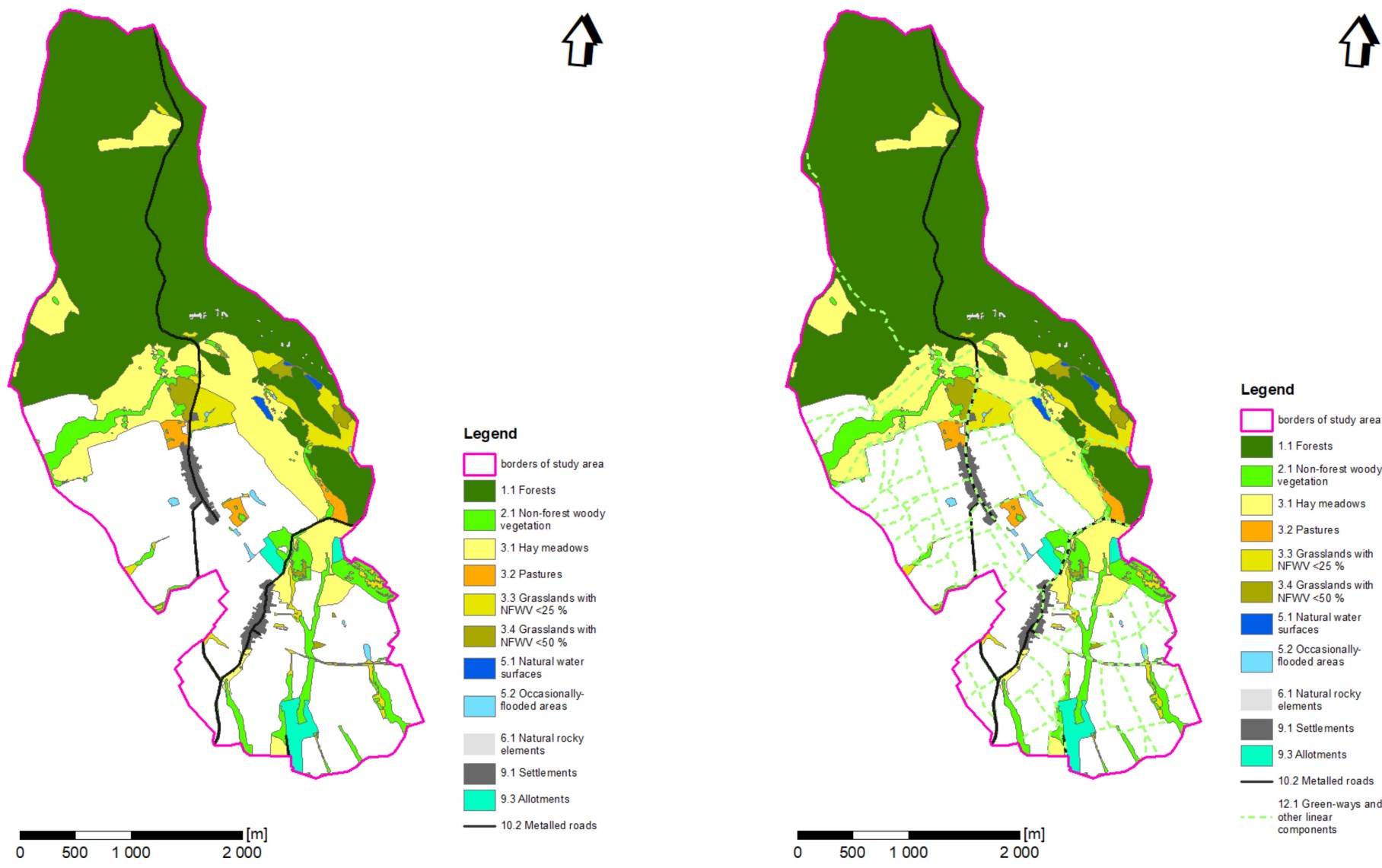

**Figure 12.** The map of connectivity of green infrastructure: current (on **left**), after application of proposals (on **right**).

## 4. Discussion

Greenways and other linear components of the green infrastructure are important tools in landscape planning and land use optimisation. In line with the principles of landscape planning, with this paper we contribute to the demonstration of innovative methods for greenways planning, which by applying ecostabilization principles can help to make decisions about how to place new objects and facilities such as greenways most appropriately in the landscape. The planning of greenways on the basis of ecological analysis of the landscape can be incorporated into spatial planning documents or rural development programmes.

Our greenways planning approach uses the steps of landscape ecological planning and planning of territorial systems of ecological stability, the latter of which is commonly carried out in Slovakia and the Czech Republic. Similar steps are used in research from countries such as the USA [19]. Other research approaches, using GIS and modern tools with Multiple Criteria Decision Analysis, provide useful tools for decision-making frameworks, and are more useful for policy makers and different stakeholders seeking to implement greenways [32]. Another study focuses on green infrastructure as a compact system, where greenways are represented as linear components of green infrastructure in a rural agricultural area, and evaluates the connectivity of green infrastructure over time [33].

The main contribution of our article is to apply innovative evaluations aimed at increased ecostabilization and erosion risk reduction, and using an aesthetic approach in greenways planning, with appropriate consideration of other more socio-economical functions as well. The approach described in this paper is also applicable to Slovakia's recovery and resilience plan, which (among other things) is aimed at boosting investments in sustainable transport. The limitations of this paper are a lack of decision-making tools or opportunities for stakeholders and policy makers, and also a lack of scenario design of the sort which could be used for advanced decision-making.

Using the method of the superposition of land cover and relief configuration maps, we obtained important data about land use, including potential erosion and visual impact; macrostructure classification data was also used in determining visual impact. These diagnostic steps helped us to define our problems and find solutions in terms of sustainable landscape planning. Sustainable landscape planning involves eliminating environmental problems and balancing socio-economic development with natural resources in the target area [34]. In Portugal, greenways planning used a landscape eco-cultural analysis consisting of biophysical analysis (geology, soil, hypsometry/slopes, sun exposure, landscape morphology, geomorphology, biogeography, flora), dynamic analyses (landscape geomorphological dynamics, namely morphogenesis and soil rate), vegetation analysis (actual vegetation, potential vegetation, habitat according to the Habitat Directive) and cultural analysis (landscape cultural elements) [21]. In comparison, the Slovak Republic has yet to apply greenways planning; instead, work has focused on identification and protection of ecological networks embodied in territorial systems of ecological stability, the delineation of which include complex analysis of each landscape's various components and the relevant abiotic, biotic, and socio-economic (positive and negative) factors. For both approaches, the most important analyses are those of the current landscape structure [35]. However, in order to increase ecological stability and fulfil the recreational potential, it is necessary to apply existing methods which can also provide necessary tools for decision-making regarding the creation of greenways or ecological network elements.

Our approach helped us to specify the number, dimensions and location of ecostabilisation elements whose incorporation into the landscape we propose. Ecological stability evaluation is one of the tools for the appraisal of conditions, and a requirement for adding elements of ecological networks to the landscape [28].

For communities, greenways bring an opportunity for recreation; their use has conditional and health benefits; they offer alternative transportation links, provide habitat and biodiversity protection and economic development; and they also have aesthetic, visual, and psychological benefits. The involvement of the local population in recreational

activities such as biking increases the importance of having greenways available for active use [36]. The proposed greenways route in our study area should also have the side-effects of making the landscape accessible, supporting the development of agrotourism, as well as tourism and ecotourism development in the local communities Nižná Pokoradz and Vyšná Pokoradz, and strengthening the connectivity of rural communities with each other and with the town. An attractive landscape with greenways may motivate inhabitants and tourists to use ecological transport and adopt healthy lifestyles and to act in ways that prompt local economic development. According to Domon [37], greenways by their nature provide more opportunities for stimulating rural development than do those landscapes which were designed, or evolved, without consideration of the connectivity between cultural or natural objects. Greenways are generally based on local pathways, which allow more optimal use of the natural and cultural resources in the landscape.

The map of potential erosion provides very important information for decision-making in landscape planning and design. Regarding vegetation, one highly desirable measure would be the planting of vegetation to form bio-corridors, interactive elements, or erosion-preventative vegetation fringes, in order to provide protection where there is medium and high soil erosion risk, particularly in the agricultural landscape. The potential erosion risks can also be decreased by erosion-protective management measures (erosion-protective distribution of parcels and crops, re-consolidation of field blocks to reach protective size and form, etc.), as well as other agrotechnical measures which protect against soil erosion. For better decision-making, it is necessary to use calculation of real soil loss erosion using RUSLE models in order to get a more precise evaluation.

Our proposals are in line with spatial planning goals concerned with land protection, land use and landscape design. At the same time, our proposals are also compatible with the program of economic and social development of Rimavská Sobota for the period of 2014–2020, in that building cycling routes will strengthen tourist development and provide leisure-time and sport opportunities for inhabitants.

The elaborated documents and proposals are potentially useful in spatial and landscape planning for simplified or complex land consolidation projects, support for territorial systems of ecological stability and rural development programmes, and plans for economic and social development. Finally, greenways benefit various NGOs and local action groups focusing on landscape protection, landscaping, tourism development or rural development.

The selected study area of the cadastres of Nižná Pokoradz and Vyšná Pokoradz has recreational potential due to their location between two geomorphologically-different areas, which is reflected in the dynamic relief of the area. The dominant landscape types are forests on uplands and large-sized blocks of arable land on flat relief. Grasslands are also important, especially pastures and hay meadows, grasslands with non-forest woody vegetation, and grasslands as part of mosaics of various landscape elements. All of the natural values of the study area coexist in harmony with cultural heritage, providing the potential for tourist development, particularly ecotourism.

This area has medium ecological stability and is affected by multiple negative elements and phenomena, from which arise the necessity to introduce eco-stabilisation elements into problematic localities. The planting of bio-corridors or protection of other bio-centres has never been realised within the local territorial system of ecological stability, and the only protection is provided in protected areas–the SCI Pokoradzské Jazierka and SPA Cerová Vrchovina-Porimavie. The local territorial system of ecological stability is also not relevant in terms of landscape and vegetation dynamics. For recreational land use, the most important location is Kamenný Janko (a panoramic viewpoint), but there is no official tourist or cycling route connecting the town, rural settlements, and recreational interests directly. For that reason, we have proposed a main greenway containing a loop which connects these places.

In order to reach the goal of synergy with management plans for the existing protected areas SCI and SPA, it is necessary to integrate those plans into the planning of greenways and other linear elements in order to prevent negative effects on objects of conservation.

The management plans of SPA Cerová Vrchovina-Porimavie propose the planting of linear woody vegetation with mixed shrubs and trees, solitaire trees, and patches of woody vegetation, and the greening of arable lands. When planning greenways on existing unmetalled roads through the south edge of SCI Pokoradzské Jazierka, it is necessary to secure permission from the relevant environmental authorities. The management plan does not support the creation of greenways or cycle routes, but proposes drawing up project documentation for educational routes through protected areas for optimal visitor mobility.

The most important factors are the opinions of, and possibilities of collaboration with, the local stakeholders (landowners, land users, local farmers, NGOs, micro-regional authorities, local action groups, and the municipality) who have a key role in decision-making and realisation of plans.

The methodological approach described in this paper has been presented as a basis for the planning of greenways and other supporting linear green infrastructure. Such a plan can be applied in landscape or spatial planning. The proposed greenways routes for our study area would connect with existing local road and field pathways (including red-signed hiking routes and other field roads), and link together the town, nearby allotments, two rural communities with socio-economically valuable characteristics, and important natural and recreational spaces. They would also provide an opportunity to connect to the neighbouring villages. At present, designing greenways routes could be of benefit not only for local inhabitants, but also for tourists from remote parts of the regions, or from outside the region, who can visit the area via the tourist route of Mária Széchy.

## 5. Conclusions

Greenways as a linear component of green infrastructure are a good example of the modern environmental concept of recreational landscape utilization and sustainable mobility. In order to demonstrate this definition, we performed analyses and evaluation of size categories of agricultural cultures, ecological stability, connectivity, potential erosion, real erosion forms, and certain socio-economic phenomena selected according to recreational potential. Using these analyses and evaluations, we have proposed a network of linear components of green infrastructure (main greenways, supportive greenways and other linear elements), and have demonstrated the potential benefits of these linear components for the landscape.

By applying these proposals, we could eliminate negative visual impact by delimiting macrostructures, increase ecological stability of the study area, provide more connectivity (which would also assist migration of various animals), decrease potential erosion risks, and stabilize real forms of erosion. By planting linear woody vegetation, we would support the important habitats of several birds, including birds of conservational interest in the spatial protected area Cerová Vrchovina-Porimavie.

Greenways planning can provide a synergistic combination of sustainable rural development, landscape protection, and sustainable tourism, which can help to provide optimal landscape utilization and provision of multiple ecosystem services. Greenways as part of green infrastructure may contribute to climate change mitigation. Greenways linked to a network of hiking and biking trails contribute to improving the quality of tourist infrastructure and may increase the numbers of tourist visitors in the study area. Finally, greenways create an alternative rural connection of settlements and have social value by providing sustainable mobility to work, school and recreation, as well as by their potential to increase physical activity, health, and well-being.

**Author Contributions:** Conceptualization, methodology, formal analysis, investigation, visualisation, and writing (original draft preparation)–J.M.; supervision–J.Š. All authors have read and agreed to the published version of the manuscript.

**Funding:** This research was funded by Slovak Research Agency of the Ministry of Education, Science, Research and Sport of the Slovak Republic, VEGA grant number 2/0135/22: Research of specific landscape elements of bio-cultural landscape in Slovakia.

**Institutional Review Board Statement:** Not applicable.

**Informed Consent Statement:** Not applicable.

**Data Availability Statement:** Publicly available datasets were analyzed in this study. These data can be found here: [https://www.shmu.sk/sk/?page=1&id=klimat_zrazkosiet&region=BB], accessed on 29 November 2021; [https://portal.vupop.sk/portal/apps/webappviewer/index.html?id=d89cff7c70424117ae01ddba7499d3ad], accessed on 29 November 2021; [https://gis.nlcsk.org/arcgis/rest/services/Inspire/PodneTypy/MapServer], accessed on 29 November 2021; [https://zbgis.skgeodesy.sk/mkzbgis/sk/teren/export], accessed on 29 November 2021; [https://zbgis.skgeodesy.sk/mkzbgis/sk/teren/export], accessed on 29 November 2021; [https://gsaa.mpsr.sk/2021], accessed on 29 November 2021; [https://www.geoportal.sk/sk/sluz-by/mapove-sluzby/wmts/wmts-zbgis.html], accessed on 29 November 2021; [http://www.pamiatky.sk/po/po?Kraj=6&Okres=51&Obec=1620&KatastralneUzemie=2139&Uli-ca=&OrientacneCislo=&UnifikovanyNazovPO=&CUZText=&SearchBut-ton=H%C4%BEada%C5%A5], accessed on 29 November 2021; [http://maps.sopsr.sk/], accessed on 29 November 2021; [http://www.sopsr.sk/ps.chvu2/files/Cerova-vrchovina-Porimavie.pdf], accessed on 29 November 2021; [http://www.sopsr.sk/poprpokoradzskejazierka], accessed on 29 November 2021; [https://mapy.dennikn.sk], accessed on 29 November 2021; [https://mapy.tuzvo.sk/HOFM], accessed on 29 November 2021.

**Acknowledgments:** We are grateful to Mathew Sebastian and James Asher for English proofreading.

**Conflicts of Interest:** The authors declare no conflict of interest.

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
