# Peer review of "Application of Landscape-Ecological Approach for Greenways Planning in Rural Agricultural Landscape"

_environments, doi:10.3390/environments9020030_

Round 1

Reviewer 1 Report

This paper proposed a research framework to design the greenways in the study area lies in south from Central Slovakia. The topic is interesting and the work is complete. However, this paper is more like a planning project than a research. The greenway planning is worked out by a serious of trivial steps which is weak in systematicness. The systematic and theoretical property of this paper needs to be improved.

There are some specific comments below.

(1) There are seven keywords in this paper. The number of keywords should reduce.

(2) The concept, characteristics and functions of greenways could be introduced more briefly in the introduction section.

(3) A review of the greenways planning approaches is needed in the paper.

(4) The data source and processing of the paper should be introduced uniformly in front of the methods.

(5) In the discussion section, the approaches in this paper should be compared with other approaches. And the strengths and limitations of the research should be discussed.

Author Response

Dear reviewer,

thank you very much for your appreciation and recommendation. We tried to adjust the paper according to the requirements of all of the reviewers.

Comments and Suggestions for Authors:

This paper proposed a research framework to design the greenways in the study area lies in south from Central Slovakia. The topic is interesting and the work is complete. However, this paper is more like a planning project than a research. The greenway planning is worked out by a serious of trivial steps which is weak in systematicness. The systematic and theoretical property of this paper needs to be improved.

Thank you very much again for your comments and suggestions. We tried to use many of them to improve our article. We used systematic represented by steps typical for Landscape ecological planning and Terrain systems of ecological stability which are basically used in Czech Republic and Slovakia, and we also compared it in following revised version with other studies respectivelly approaches. We also improved theoretical property in this paper.

There are some specific comments below:

  • There are seven keywords in this paper. The number of keywords should reduce.

  • The concept, characteristics and functions of greenways could be introduced more briefly in the introduction section.

I reduced paragraphs in introduction section, I think they are shorter now.

  • A review of the greenways planning approaches is needed in the paper.

I added paragraph with an overview of the greenways planning aproaches into introduction.

  • The data source and processing of the paper should be introduced uniformly in front of the methods.
  • In the discussion section, the approaches in this paper should be compared with other approaches. And the strengths and limitations of the research should be discussed.

The discussion section was improved by comparison of other approaches and also with strengths and limitations of research.

Best regards,

Jakub Melicher

Reviewer 2 Report

I find the classification of the Greenways very interesting and imaginative but effective...optimal in the conclusions
It is important, what the authors have done, to over consider and define greenways as recreational landscape utilization and sustainable mobility in scientific and cartographic terms.

Author Response

Reviewer #2
Manuscript Number: environments-1555281
Title: Application of landscape-ecological approach for greenways planning in rural agricultural landscape
Type: Article

Dear reviewer,

thank you very much for your appreciation and recommendation. We tried to adjust the paper according to the requirements of all reviewers.

Comments and Suggestions for Authors:

I find the classification of the Greenways very interesting and imaginative but effective...optimal in the conclusions. It is important, what the authors have done, to over consider and define greenways as recreational landscape utilization and sustainable mobility in scientific and cartographic terms.

Thank you very much for Comments and Suggestions. We considered and updated revised version of following paper by more suggestions and comments of all reviewers.

Best regards,

Jakub Melicher

Reviewer 3 Report

I really enjoyed reading his manuscript. I would suggest minor corrections.

  1. Language needs to be improved as can be seen on line 14 and 23.
  2. First paragraph lines 29-38 came rather too early, it should have come after sufficient background is laid on the issue and particularly with global examples.  
  3. Line 225 table 4- can you covert ha to km2.
  4. I like the depth of analysis and discussion can be improved further.

Author Response

Reviewer #3:
Manuscript Number: environments-1555281
Title: Application of landscape-ecological approach for greenways planning in rural agricultural landscape
Type: Article

Dear reviewer,

thank you very much for your appreciation and recommendation. We tried to adjust the paper according to the requirements of all of the reviewers.

Comments and Suggestions for Authors:

I really enjoyed reading his manuscript. I would suggest minor corrections.

Thank you again very much for positive review. I answered all minor corrections below.

  1. Language needs to be improved as can be seen on line 14 and 23.

I improved language following recommendation. English was revised by native English speaker.

  1. First paragraph lines 29-38 came rather too early, it should have come after sufficient background is laid on the issue and particularly with global examples.  

I put global data represented by EU scale in the first paragraph of the introduction.

  1. Line 225 table 4- can you covert ha to km2.

I would rather remain ha due to uniform use of the format in this paper (also used in Table 3, Figure 5, page 10, 14, 18)

  1. I like the depth of analysis and discussion can be improved further.

I revised discussion and added other paragraph and ideas that improved discussion.

Best regards,

Jakub Melicher

Round 2

Reviewer 1 Report

The English is clumsy and some sentences have grammatical mistakes.

Author Response

Reviewer #1
Manuscript Number: environments-1555281
Title: Application of landscape-ecological approach for greenways planning in rural agricultural landscape (2nd round)
Type: Article

Dear reviewer,

thank you very much for your appreciation and recommendation. The extensive editing of English language and style was realized by two different native speakers so we hope that revised version of manuscript might satisfy your requirement.

Best regards,

Jakub Melicher